# Physics of Language Models: Part 3.2, Knowledge Manipulation

## [Extended abstract]*

**Zeyuan Allen-Zhu**
FAIR at Meta
`zeyuanallenzhu@meta.com`

**Yuanzhi Li**
Mohamed bin Zayed University of AI
`Yuanzhi.Li@mbzuai.ac.ae`

## Abstract

Language models can store vast factual knowledge, yet their ability to flexibly use this knowledge for downstream tasks (e.g., via instruction finetuning) remains questionable. This paper investigates four fundamental knowledge manipulation tasks: **retrieval** (e.g., "What is person A's attribute X?"), **classification** (e.g., "Is A's attribute X even or odd?"), **comparison** (e.g., "Is A greater than B in attribute X?"), and **inverse search** (e.g., "Which person's attribute X equals T?").

We show that language models excel in knowledge retrieval but struggle even in the simplest classification or comparison tasks unless Chain of Thoughts (CoTs) are employed during both training and inference. Moreover, their performance in inverse knowledge search is virtually 0%, regardless of the prompts. Our primary contribution is a *controlled, synthetic experiment* that confirms these weaknesses are *inherent* to language models: they cannot efficiently manipulate knowledge from pre-training data, even when such knowledge is perfectly stored in the models, despite adequate training and sufficient model size. Our findings also apply to modern pretrained language models such as GPT-4/4o, thus giving rise to many Turing tests to distinguish Humans from contemporary AIs.

## 1 Introduction

Knowledge is a fundamental component of human civilization and intelligence. Throughout our lives, we accumulate a vast amount of knowledge and learn to use it flexibly. Large language models like GPT-4 (OpenAI, 2023; Bubeck et al., 2023) have demonstrated an impressive capacity to memorize knowledge, arguably surpassing any human. These models also show signs of being able to manipulate this knowledge to solve various problems, arguably reaching an L2 or L3-level of intelligence (Allen-Zhu & Xu, 2025).

In this work, we aim to understand how transformer-based language models manipulate the knowledge they have memorized during pretraining and use it flexibly to solve different tasks at inference time. For example, can language models determine if a person's college is ranked higher than another one's based on its stored 2023 US News university ranking knowledge? Can they answer questions such as "Was Joe Biden born in an odd year?" or "Was Donald Trump born earlier than Nancy Pelosi?" based on their memorization of celebrities' birthdays?

Spoiler: NO! Even the strongest models, GPT-4o and Llama-3.1-405B, *still* fail as of October 1, 2024 (ICLR submission date; see Figure 5). This paper explains *why* these failures occur. We have

---

*The first six papers in the *Physics of Language Models* series were presented as a two-hour tutorial at ICML 2024 in Austria (`youtu.be/yBL7J0kgldU`). A one-hour deep dive into Parts 3.1 and 3.2 is available at `youtu.be/YSHzKmEianc`. Full and future editions of Part 3.2, including additional experiments and potential code releases, can be found at `physics.allen-zhu.com` and `ssrn.com/abstract=5250621`.

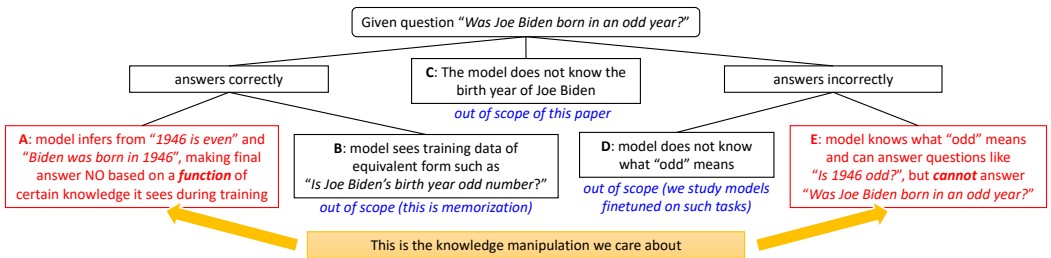

Figure 1: We study (A) vs (E) as knowledge manipulation. With a pre-trained model over internet data, it is very hard to determine whether (B,C,D) has happened due to the uncontrollability of internet data.

verified that the same counterexamples continue to hold in newer models, such as Gemini 2.0 and Claude 3.5/3.7, and may include these findings in future versions of this write-up.

In other words, we are interested in questions that are *functions* of specific knowledge from the pretraining data, and study a language model's ability to answer questions during inference time. Knowledge manipulation is arguably *a simplest form of logical reasoning*. To answer questions like "Is Person A's attribute X good?", a model not previously exposed to this sentence in its training data may draw conclusions from other data such as "Person A's attribute X equals T" and "T is good".

In this paper, "knowledge" refers to *factual knowledge* (e.g., knowledge graph), and we explore whether a language model can logically manipulate such knowledge embedded in its model weights. Other research may focus on in-context knowledge or RAG (Lewis et al., 2020; Cai et al., 2022; Liu et al., 2020; Jiang et al., 2023b; Mao et al., 2020; Parvez et al., 2021; Komeili et al., 2021; Ram et al., 2023; Siriwardhana et al., 2023), where the model responds to queries about a *provided paragraph* in the context (possibly via RAG).

Extensive research has been conducted on the question-answering capabilities of language models at inference time (Sun et al., 2023; Singhal et al., 2022; Omar et al., 2023; Hernandez et al., 2023; Richardson & Sabharwal, 2020; Peng et al., 2022; Petroni et al., 2019; Naseem et al., 2021), primarily focusing on models trained with internet data. A significant challenge in determining whether these models can manipulate knowledge is to ascertain if the internet data already contains the exact or equivalent question, or if the models genuinely performed logical deduction during inference time.

We are particularly interested in scenarios *without data contamination*: the questions or their equivalent forms should not appear in the model's training data, while the same "function" for other knowledge should be present — thus ensuring the model understands the function. For example, can the model determine "Was Joe Biden born in an odd year?" if it has not encountered this sentence or its equivalents during pretraining (such as "Is Joe Biden's birth year divisible by 2"), but can infer from "Biden was born in 1942" and "1942 is not odd"? Answering such questions requires the model to both memorize and comprehend the knowledge. (See Figure 1.)

To address the *unpredictability of internet data*, Allen-Zhu & Li (2024; 2025) developed synthetic pretrain data containing controlled biographies for up to $N = 20$ million individuals. They explored how a language model stores and extracts knowledge about these individuals after-pretraining. Here is an example of their biography data:

> Anya Briar Forger was born on October 2, 1996. She spent her early years in Princeton, NJ. She received mentorship and guidance from faculty members at Massachusetts Institute of Technology. She completed her education with a focus on Communications. She had a professional role at Meta Platforms. She was employed in Menlo Park, CA.

(1.1)

Allen-Zhu & Li (2024) found that a pretrained model may struggle to *extract* stored knowledge from biographical data unless the data is sufficiently *knowledge-augmented*, meaning the same biography has diverse and well-permuted English descriptions. This augmentation aids in accurately answering extraction queries such as "Which city was Anya Briar Forger born in?" While we recommend reading our concurrent work (Allen-Zhu & Li, 2024) first, this paper can be read independently.

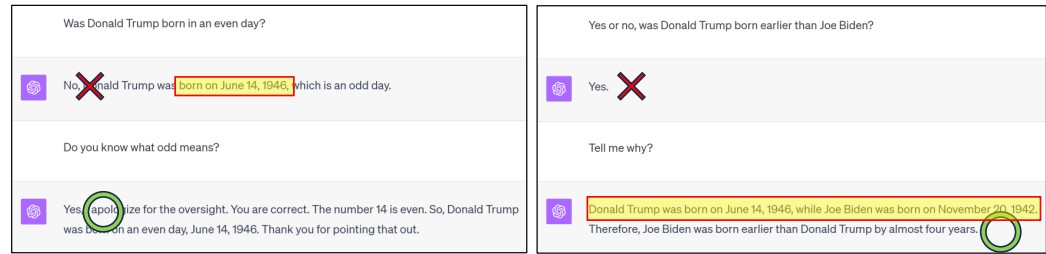

Figure 2: GPT-4 struggles to answer simple knowledge manipulation questions; but when CoT is used, where the person's attributes are first explicitly spelled out, GPT-4 can correctly answer them. More GPT-4 examples are in Figure 3, 4, and the full paper. When we prepared this paper we used GPT-4 of 2023. As of Oct 1, 2024 (ICLR submission date), these counterexamples still hold for GPT-4o and Llama-3.1-405B (see Figure 5). Future versions will expand on these with additional counterexamples for Claude 3.5, Gemini 2.0, and possibly more.

## 1.1 OUR METHODOLOGY AND RESULTS

This paper further explores whether a model, pre-trained on augmented biography data, can *manipulate* its knowledge after instruction finetuning. We investigate its ability to handle queries that require reasoning about personal attributes, such as "Was Anya born in a southern city?" or "Is Anya's university better than Sabrina's?"

During training, the model learns from the biographies of all $N$ individuals and the knowledge manipulation question-answer (QA) texts from a subset of individuals (the in-distribution set $\mathcal{P}_{\text{train}}$). We evaluate the model's *out-of-distribution* (OOD) generation accuracy by testing it on the remaining subset (the out-of-distribution set $\mathcal{P}_{\text{test}}$), where it has seen the biographies but not the QAs during training. Including $\mathcal{P}_{\text{train}}$ in the training data ensures the model encounters enough examples to comprehend the QAs. We focus on the model's OOD accuracy on $\mathcal{P}_{\text{test}}$, reflecting its true capability in logical deduction during inference time, as opposed to on $\mathcal{P}_{\text{train}}$ which could easily reach 100%.

We study four basic types of knowledge manipulations: retrieval, classification, comparison, and inverse search, which cover most real-world scenarios.[1]

KNOWLEDGE RETRIEVAL. Extending work on knowledge extraction (Allen-Zhu & Li, 2024), we finetune the model to retrieve (1) part of an attribute or (2) multiple attributes at once. We discover a model may

- correctly answer "What is the birth date of Anya" as "June 27th, 1997", but struggle with "What is the birth year of Anya" (**Result 2**); and
- correctly answer "Which company and where did Anya work" but fail on "Where and which company did Anya work." (**Result 1**)

These serve as **preliminary evidence** suggesting the necessity of a Chain-of-Thought (CoT) for knowledge manipulation. The model must *explicitly state* the birth month/day to deduce the birth year, or *explicitly state* the company name before the work city location.

KNOWLEDGE CLASSIFICATION. We finetune the model for classification tasks on its stored knowledge; for instance, "What degree did Anya receive?" may require ternary classification (art, science, engineering) based on her major. Language models often struggle with such tasks unless they (1) generate answers in CoT manner or (2) are finetuned with a significantly larger number of samples than theoretically necessary.

Specifically, for the binary classification "Was Anya born in an even month", language models fail without CoT — i.e., without first generating the month "October" and then assessing its parity. This remains true even if the model is *sufficiently* trained

---

[1]One could also explore combinations, such as "Is A's wife's university ranked higher than B's?" or "Is the person born on June 27th, 1997, and studied at MIT named with an initial A?" These would further complicate the tasks. Given that we show mostly negative results, focusing on the basic forms suffices.

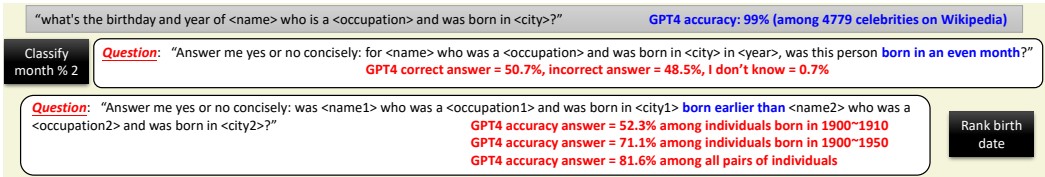

Figure 3: Knowledge classification and ranking on WikiBio using GPT-4. Details are in the full paper.

- to answer everyone's birth month with 100% accuracy,
- on 25,000 QA samples, more than needed to classify 12 months to 2 classes,

This reveals that language models cannot efficiently be trained+finetuned to perform **even a single step of knowledge manipulation** during inference time without CoT (**Result 3**). Furthermore, our findings reveal:

- Including sufficient CoT samples in training does not enhance non-CoT inference (**Result 4**);
- Improving model's knowledge extraction don't improve its manipulation ability (**Result 5**).

**Importantly**, this is different from and do not contradict to most common CoTs used in practice at enhancing math or reasoning skills; for example, GPT-4 can skip a computation step and answer whether the sum of $a$ and $b$ is even for $a, b \in [12]$, without writing down their sum explicitly. More broadly, many *in-context* reasoning can be done mentally without writing down (Ye et al., 2025a).

KNOWLEDGE COMPARISON. This task involves determining if one attribute is greater than another, based on a predefined ranking. For instance, "Is Anya's university better than Sabrina's?" requires a Yes/No response based on the universities' rankings. Our results align with those from the classification case: models struggle to perform knowledge comparisons effectively without CoTs. For instance, the accuracy of comparing knowledge among 100 options is barely random guess, even with $2,500,000$ training samples, more than enough to learn to rank 100 objects (**Result 3-5**).

KNOWLEDGE INVERSE SEARCH. This involves identifying a person based on their attributes, such as "Who was born on October 2, 1996 in Princeton..." when the knowledge is only forwardly presented in the training data: "Anya Forger was born on October 2, 1996..." We discover that language models **cannot perform this task**, regardless of training methods, data, or model size, unless the knowledge is already presented inversely in the data (**Result 8**).[2] This suggests that *language models cannot be used as databases*.

*Remark* 1.1. Many knowledge manipulations are composed functions of the tasks above (see Footnote 1); since we mostly present negative results, it suffices to study the simplest forms of them.

IN PRACTICE. We demonstrate that modern large models like GPT-4/4o and Llama-3 (see Figure 2, 3, 4, 5) also struggle with these tasks (**Result 6, 8**). Future editions of this paper will include additional counterexamples for Gemini 2.0, Claude 3.5, and possibly more, suggesting these limitations may be *inherent and universal* to generative language models — and *not easily overcome by scaling*.

## 1.2 OUR CONTRIBUTIONS

We discover that language models, through controlled experiments and pre-trained on synthetic data, perform poorly at basic knowledge manipulation tasks. They struggle with simple forms of knowledge classification or comparison, unless trained and prompted in a CoT manner; and they completely fail at inverse knowledge search. This synthetic setting acts as a *simple, yet important testbed* for future studies to enhance in language models' knowledge manipulation abilities.

**Connection to prior work on CoTs.** The formal introduction of CoT (Wei et al., 2022) and subsequent studies have highlighted the significance of CoTs for complex in-context computations,

---

[2]A concurrent study (Berglund et al., 2023) observed similar results, and called this "reversal curse."

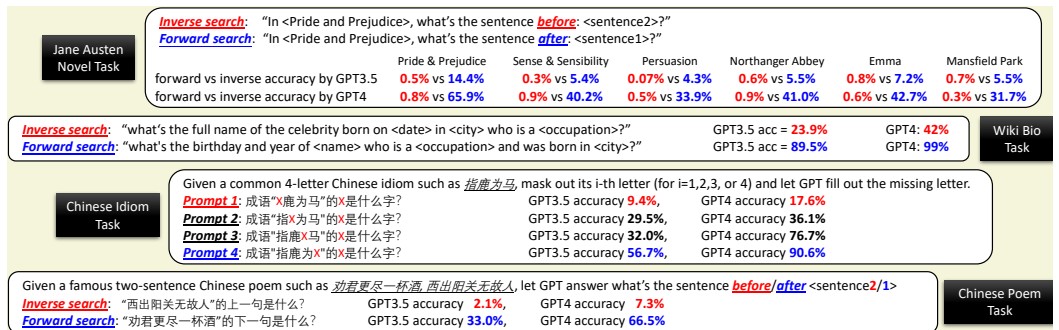

Figure 4: Forward search vs inverse search on ChatGPT (GPT3.5 / GPT-4); details in the full paper. (While inverse search may seem challenging even for humans, we have designed the Chinese idiom/poem tasks that are allegedly simple for many high school graduates in Chinese education.)

such as solving math problems. Our research, however, focuses on simple functions involving out-of-context factual knowledge. For instance, GPT-4 can accurately answer "Is the sum of $a$ and $b$ an even number?" (for $a, b \in [12]$) without explicitly calculating $a + b$.

Their paper also touched knowledge manipulation questions, such as "Did Aristotle use a laptop?" or "Would a pear sink in water?" from the StrategyQA dataset Geva et al. (2021). Although GPT-4 can answer some of these Yes/No questions today, it is unclear if this is due to data contamination or an inherent ability to manipulate knowledge without CoTs. Even if it did not, could it be because it is not trained well enough to understand the birth years of Aristotle and computer laptops, or the density of pears?

This underscores the need for controlled, synthetic experiments to eliminate such possibilities and discover the language model's true capabilities on knowledge manipulation tasks (see Figure 1 again). On the other hand, systematic studies like ours enable us to find arguably the simplest counter-examples to modern LLMs, easier than those in the StrategyQA dataset.

**Connection to humans.** Our findings suggest a Turing test to distinguish humans from modern generative language models (at least as of today). Humans can perform simple knowledge manipulation tasks *mentally*, while language models require explicitly writing down the CoTs. Despite the challenge of inverse search for humans, we identified tasks easily solvable by humans but not by GPT-4 (refer to Figure 4). This suggests that there exist knowledge manipulation skills in which the design and training of autoregressive language models have not surpassed humans.

**Connection to industry.** While this paper reveals that novel techniques are needed to fundamentally improve a language model's knowledge manipulation ability, immediate mitigations are also possible. This includes generating more CoT data (**Results 3-5**) and employing methods like retrieval augmented generation (RAG) (Lewis et al., 2020) and reversal training (Golovneva et al., 2024; Nguyen et al., 2024; Guo et al., 2024) to help inverse search, or multi-token prediction (Gloeckle et al., 2024) to help partial retrieval. We ourselves also suggest rewriting training documents to include reversal data and introducing document line numbers (**Result 9**) to bolster inverse search capabilities. These strategies could inform the development of future industrial-scale language models.

## 2 MAIN BODY OF THIS PAPER

Technical details are omitted in this ICLR version to encourage readers to refer to our full paper at ssrn.com/abstract=5250621, which will also feature up-to-date experiments on this topic. We remark that the full paper underwent the ICLR 2025 review process, but we elected to present this camera-ready version as an *extended abstract*, aligning with the tradition in the theory community.

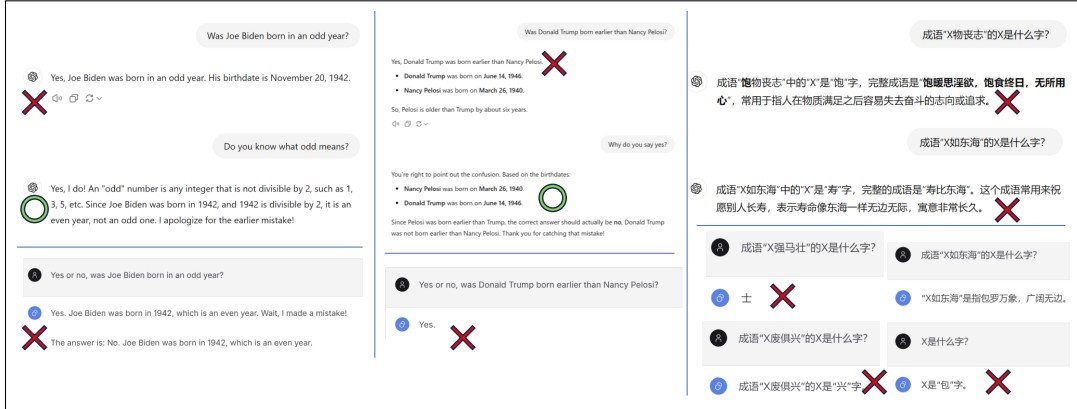

Figure 5: Even as of Oct 1, 2024, GPT-4o (top) and Llama-3.1-405B (bottom) still fail on simple knowledge classification (left), knowledge comparison (middle) and inverse search (right) tasks.

## 3 CONCLUSION

We use *controlled experiments* to show some fundamental limitation of language models to manipulate knowledge during inference time *even under the strongest pretraining setting, regardless of model size, data size, etc*. Our work sheds light on why extremely large language models like GPT-4 are still bad at even the simplest, single-step knowledge manipulation, and give surprisingly simple such counter-examples (see Figure 2, Figure 5). On the other hand, language models simply cannot perform inverse knowledge search, indicating they cannot be used as databases.

While this paper reveals that novel techniques are needed to fundamentally improve a language model's knowledge manipulation ability, immediate mitigations are also possible. This includes generating more CoT data (Results 3-5) and employing methods like retrieval augmented generation (RAG) (Lewis et al., 2020) and reversal training (Golovneva et al., 2024; Nguyen et al., 2024; Guo et al., 2024) to help inverse search, or multi-token prediction (Gloeckle et al., 2024) to help partial retrieval. We ourselves also suggest rewriting training documents to include reversal data (Result 9) and introducing document line numbers (Result 9) to bolster inverse search capabilities. These strategies could inform the development of future industrial-scale language models.

Finally, Part 3 of this work series focuses on how language models store, extract and manipulate knowledge (including Part 3.1 and 3.3 (Allen-Zhu & Li, 2024; 2025)). We also cover grade-school math reasoning in Part 2 (Ye et al., 2025a;b), hierarchial language structure learning in Part 1 (Allen-Zhu & Li, 2023), and architecture design in Part 4 (Allen-Zhu, 2025).

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
