# OpenReview forum: "Physics of Language Models: Part 3.2, Knowledge Manipulation"
_ICLR.cc/2025/Conference — ICLR 2025 Poster_

### Official Review · Reviewer_Zspe · 2024-10-18

**Soundness:** 3
**Presentation:** 4
**Contribution:** 3
**Rating:** 6
**Confidence:** 4

**Summary:**

This paper investigates the ability of large language models (LLMs) to manipulate stored factual knowledge through various tasks such as retrieval, classification, comparison, and inverse search. The authors argue that while LLMs like GPT-4 excel in knowledge retrieval, they struggle with classification and comparison tasks unless trained and tested with Chain of Thought (CoT) reasoning. Inverse knowledge search is highlighted as a critical limitation, with performance remaining near-zero across different datasets, architectures, and model sizes. The paper utilizes synthetic biographical datasets to conduct controlled experiments, confirming these weakness of LLMs and revealing the need for enhanced training strategies to improve LLMs’ knowledge manipulation abilities.

**Strengths:**

* This paper focus on knowledge manipulation addresses a vital challenge in NLP, with implications for retrieval-based tasks.
* The paper Introduce inverse knowledge search as an evaluation criterion adds a new dimension to LLM evaluation.
* The synthetic biography datasets and the four well-designed tasks are valuable assets for future researchers.
* The paper is well-organized and presented clearly.
* The paper offers detailed analyses and experimental results, contributing meaningfully to understanding LLM limitations.

**Weaknesses:**

Recent studies have extensively explored the challenges of LLMs in reasoning and manipulating knowledge from pre-training data. It would strengthen the paper if the authors discussed potential solutions based on their findings.

**Questions:**

None

---

> ### Author Response · Authors · 2024-11-28
> **Response to Reviewer Zspe**
>
> We sincerely thank the reviewer for carefully reading the paper and liking our presentation. We also agree that there are extensive recent studies to address LLM’s capability limitations — and our paper is actually one of the first such papers (based on arxiv date), and also quite different from most others because we do controlled experiments.
>
> We are exploring the solutions actively. There are follow-ups (including our own) that tried to for instance include reversed knowledge data into the pretraining, but this is somewhat naive and it effectively brings inverse knowledge into forward knowledge. Finding a good solution may not be easy, as otherwise it won’t happen that the same manipulation errors continued to exist in modern LLMs after a year since this paper has appeared.
>
> The good news is, we are thinking deeply into what necessary (model/training/data) changes can help facilitate the model to improve its knowledge manipulation capability **intrinsically**, and do have some partial results. We are verifying them and wish to write a follow-up paper if things work out well.
>
> Nevertheless, we believe using synthetic controlled experiment to provide a testbed is an important first step, without which it can become illusive and sometimes even debatable regarding whether a model is truly capable of performing knowledge manipulation. Thanks again for your time reading our work!

---

### Official Review · Reviewer_3F4B · 2024-10-31

**Soundness:** 4
**Presentation:** 4
**Contribution:** 3
**Rating:** 8
**Confidence:** 4

**Summary:**

Pre-trained Language Models (PLMs) are widely recognized for encoding extensive knowledge during pre-training. However, their effective utilization of this knowledge remains uncertain. This paper examines knowledge manipulation capabilities across four tasks: retrieval, classification, comparison, and reverse search.

To isolate these factors, the authors emphasize the need for a controlled setting where the model’s responses reflect actual knowledge manipulation rather than simple recall of training data. Given that recent Large Language Models (LLMs) are trained on expansive internet datasets, achieving such control is challenging. Consequently, the authors re-train several models (GPT-2, LLaMA, and Mistral) on synthetic datasets, including biographical datasets (bioS, bioR) and Q&A datasets for either pretraining or fine-tuning.

Results reveal that while LLMs perform well in knowledge retrieval, they struggle with partial knowledge extraction (e.g., identifying only the birth month instead of the full birth date), knowledge comparison (e.g., determining if a birth month is even), and reverse search (e.g., identifying a person by their birth date). The authors further observe that performance improves significantly with Chain of Thought (CoT) prompting, suggesting that LLMs benefit from generating intermediate reasoning steps (e.g., “He was born in December, which is the 12th month and therefore even”).

Finally, the authors extend their study to larger production models, such as GPT-4o and LLaMA-3-405B, revealing similar limitations, underscoring this as an area for improvement in future model iterations.

**Strengths:**

- The paper is thorough and self-contained, re-explaining key concepts, which enhances readability and makes the study easy to follow.
- The writing is clear, and methodological points are frequently illustrated with examples, adding clarity to the analysis.
- The use of synthetic datasets to examine LLMs within a controlled environment is a valuable and relevant methodological choice.
- By testing multiple models, including recent versions like GPT-4o, the paper strengthens the generalizability of its findings.
- Overall, the paper addresses a crucial issue in model capabilities that warrants broader attention.

**Weaknesses:**

Figures 3, 4, and 6 are challenging to interpret, particularly due to the color coding and the distinctions between various lines (e.g., bioS multi5 + permute vs. bioS multi5 + permute + fullname) and the variations in behavior among them. Reducing the number of results displayed and focusing on a deeper analysis of selected findings could enhance clarity.

**Questions:**

1) Could the reliance on Chain of Thought prompting be due to the fact that this approach effectively enables the model to “think” more deeply, performing additional steps or calculations that might not otherwise be possible?
2) What methods would you consider to address this limitation of LLMs?

---

> ### Author Response · Authors · 2024-11-28
> **Response to Reviewer 3F4B**
>
> We sincerely thank the reviewer for supporting this work. We can reduce the number of rows with a better focus following your suggestion. To answer your questions:
>
> **Question:**
> > Could the reliance on Chain of Thought prompting be due to the fact that this approach effectively enables the model to “think” more deeply, performing additional steps or calculations that might not otherwise be possible?
>
> **Answer:**  Adding CoT effectively divides a more complex task into simpler ones, so that the model is no longer responsible for “thinking more deeply” and thus with higher accuracy. Although we probably mean the same thing, but we interpret this as “adding CoT encourages the model not to think deeply” in some sense.
>
> **Question:**
> > What methods would you consider to address this limitation of LLMs?
>
> **Answer:**  We are exploring this actively. There are follow-ups (including our own) that tried to for instance include reversed knowledge data into the pretraining, but this is somewhat naive and it effectively brings inverse knowledge into forward knowledge. We are thinking more deeply into what necessary (model/training/data) changes can help facilitate the model to improve its knowledge manipulation capability intrinsically, and do have some partial results. We are verifying them and wish to write a follow-up paper if things work out.
>
> Nevertheless, we believe using synthetic controlled experiment to provide a testbed is an important first step, without which it can become illusive and sometimes even debatable regarding whether a model is truly capable of performing knowledge manipulation. Thanks again for your time reading our work!

---

> > ### Comment · Reviewer_3F4B · 2024-11-28
> >
> > Thank you for your detailed response to my questions.

---

> > > ### Author Response · Authors · 2024-12-03
> > >
> > > Thanks again for your support!

---

### Official Review · Reviewer_SsDr · 2024-11-04

**Soundness:** 3
**Presentation:** 3
**Contribution:** 2
**Rating:** 6
**Confidence:** 3

**Summary:**

This paper examines the knowledge manipulation capabilities of LLMs, such as knowledge retrieval, classification, comparison, as well as inverse search. Through controlled experiments on models like GPT2 and Mistral, the authors reveal substantial limitations in these models' ability to perform even basic manipulations, such as identifying a person’s birth year, unless that information is directly present in the model's training data or using advanced prompting methods like CoT.

**Strengths:**

1. The flaws of LLMs highlighted in this work are fundamental and significant. The testbed and methodology presented here could serve as a valuable benchmark for future generations of LLMs, potentially marking a clear boundary between AI systems that possess system II reasoning capabilities and those that do not.

2. The evaluation is reasonably thorough, including various types of knowledge manipulation tasks and examining both open and closed-source LLMs.

**Weaknesses:**

1. It's not a new finding that LLMs struggle to retrieve and apply stored knowledge effectively for solving reasoning tasks [1, 2]. However, the author doesn't cite these existing work.

2. The author reports a new finding that LLMs trained with CoT are not better at knowledge manipulation. However, examining the training samples (L360-362) reveals that the CoT format used is incomplete as it includes only intermediate answers without the full reasoning chain, which may have hindered the effect.

3. The author trains the models on synthetic biography data. It's well motivated as it allows the authors to assess knowledge manipulation without data contamination risks. However, it’s uncertain whether fine-tuning on this synthetic data might lead to catastrophic forgetting in the models, affecting their understanding of concepts like "even" or their general manipulation capabilities.

[1]  Kazemnejad, Amirhossein, et al. "Measuring the Knowledge Acquisition-Utilization Gap in Pretrained Language Models." The 2023 Conference on Empirical Methods in Natural Language Processing.

[2] Wu, Weiqi, et al. "Do PLMs Know and Understand Ontological Knowledge?." Proceedings of the 61st Annual Meeting of the Association for Computational Linguistics (Volume 1: Long Papers). 2023.

**Questions:**

1. I think your work is also related to those work on assessing the self-consistency of LLMs.

---

> ### Author Response · Authors · 2024-11-28
> **Response to Reviewer SsDr**
>
> We sincerely thank the reviewer for the detailed comments.
>
> **Question:**
> > It's not a new finding that LLMs struggle to retrieve and apply stored knowledge effectively for solving reasoning tasks [1, 2]. However, the author doesn't cite these existing work.
>
> **Answer:**  Thanks for pointing out and we shall cite them in our next versions. [1] studied pretrained models using internet data; and as we have argued, we wish to conduct controlled experiments to investigate more rigorously. [2] is an independent work to ours according to arxiv dates, but we’d love to cite that as well.
>
> **Question:**
> > The author reports a new finding that LLMs trained with CoT are not better at knowledge manipulation. However, examining the training samples (L360-362) reveals that the CoT format used is incomplete as it includes only intermediate answers without the full reasoning chain, which may have hindered the effect.
>
> **Answer:** Thanks for pointing this out. We in fact have tried many different formats of CoT and there’s no difference in the results (including the dot dot dot mentioned in Footnote 7). Our initial thinking was that, given that we are showing “adding CoT the accuracy goes up”, then **using the “simplest possible CoT English text” would make the result even stronger**.
>
> But you’re right, in this one place, that is to show “adding CoT during training doesn’t help non-CoT testing”, then one should try more detailed CoT formats for a better result. I can assure you that there’s no major difference in the accuracies (we’ve done this experiment before the submission), so we are happy to include a footnote saying so.   Specifically, we already tried format “<name1> was born on <date1>, and <name2> was born on <date2>” and similar variants to all of our manipulation CoTs.
>
> **Question:**
> > The author trains the models on synthetic biography data. It's well motivated as it allows the authors to assess knowledge manipulation without data contamination risks. However, it’s uncertain whether fine-tuning on this synthetic data might lead to catastrophic forgetting in the models, affecting their understanding of concepts like "even" or their general manipulation capabilities.
>
> **Answer:**  This is a valid concern, and it’s also precisely why we have played with different sample sizes and especially simplest possible manipulation tasks, to ensure that if successful, the finetuning shouldn’t take long (namely, how hard is it to classify 12 months into 2 classes).
>
> If you want to completely remove the possibility of catastrophic forgetting, then you can add the BIO data and manipulation QA all together for a mixed training. We have done so in our inverse knowledge search section (see right column block of Figure 6) and there is no help. For classification and comparison, we didn’t do that for the cleanest presentation, as Figure 4 is already very complex. We can include the experiments for mixed training as well (namely, adding BIO data + manipulation data together for joint training) and the conclusion that it helps a bit, but still cannot fully resolve the problem.
>
> Thanks again for reading this paper, and we hope these have to some extent addressed your questions.

---

> > ### Comment · Reviewer_SsDr · 2024-11-29
> >
> > Thank you for your response. I'll keep my original score.

---

> > > ### Author Response · Authors · 2024-12-03
> > >
> > > Thanks again for your time! Sincerely hope this paper can get in, so we can better focus on the follow-ups.

---

### Official Review · Reviewer_afmD · 2024-11-04

**Soundness:** 2
**Presentation:** 2
**Contribution:** 2
**Rating:** 6
**Confidence:** 3

**Summary:**

This paper investigates the fundamental knowledge manipulation abilities of language models, particularly in logical reasoning tasks that require combining multiple pieces of knowledge rather than finding direct answers in training data. Using a controlled biography dataset, the study examines four key reasoning abilities—retrieval, classification, comparison, and inverse search—that broadly cover real-world scenarios. Results reveal that while models perform well in basic knowledge retrieval, they struggle with tasks requiring partial extraction or retrieving multiple attributes. Classification and comparison tasks, especially those involving attribute comparisons, are challenging without the use of Chain of Thought (CoT) techniques. Additionally, the models are particularly weak in inverse search tasks, such as identifying a person based on attribute values, underscoring the limitations of current language models in more complex knowledge manipulation.

**Strengths:**

1. The experiments are well-controlled, focusing on minimizing extraneous variables, which allows the study to provide numerous insights (Results 1-8). Unlike most research on LLM knowledge, which often lacks such control and tends to use pre-trained LLMs, this study stands out as an important contribution.

**Weaknesses:**

1. The paper includes four knowledge manipulation tasks, but the reason for putting all four together in this single paper is not entirely clear. While the performance of each individual task is shown and discussed, it would be helpful if the authors discussed the overall implications of these results, i.e., what can be concluded when combining these findings.
2. While performance for each manipulation task is discussed, there is no analysis on why the results are as they are. For instance, why can the model perform full retrieval but perform worse in partial retrieval? Why does the performance improve when CoT is used? Although it may be challenging to fully explain these observations, the paper would be more insightful if it included some hypotheses and experiments to verify these hypotheses.
3. The paper explores limited variations in querying knowledge for LLMs, relying solely on the question forms + fine-tuning. But, other query methods, such as cloze-style prompting (with next-word prediction, which is closer to the pretraining objective), or zero-shot and few-shot prompting, could also be explored. It would be beneficial to see whether these alternatives could improve performance on manipulation tasks. This should lead to more robust conclusions about the challenges language models face with knowledge manipulation tasks. If one wants to claim that something does not work, as many solutions as possible need to be tested.

**Questions:**

1. Regarding Weakness 1, what are the overall implications of the results? Could you clarify the broader conclusions/impacts that can be drawn from the current results?
2. For Weakness 2, are there any possible experiments that could provide further insights, or is this beyond the scope of the current study?
3. For Weakness 3, would conducting additional experiments with varied settings yield valuable insights, or are the current experiments sufficient to reach definitive conclusions?

---

> ### Author Response · Authors · 2024-11-28
> **Response to Reviewer afmD [1/2]**
>
> We thank the reviewer for the detailed comments and the time reviewing this work. Let us address your concerns and questions.
>
> **Question:**
> > The paper includes four knowledge manipulation tasks, but the reason for putting all four together in this single paper is not entirely clear.
>
> **Answer:** In our mind, the most basic form of factual knowledge is (key, attribute, value) tuple, and knowledge manipulation means performing additional operations on such tuples. While there are more complex manipulation tasks (such as A is B’s brother, and B is C’s mother, so A is C’s <bla>), since we are presenting many negative results, we find it **sufficient to study “the most basic form” of knowledge manipulation**. If one can’t even perform a single step of knowledge classification (such as “A is B’s roommate and B is hispanic” , then “What’s the race of A’s roommate”), then it’s also impossible to perform harder tasks that get more knowledge tuples involved. To this extent, combining two pieces of knowledge, and classification, and ranking, and inverse search, are four typical and arguably most basic tasks.
>
> **Question:**
> > While performance for each manipulation task is discussed, there is no analysis on why the results are as they are. For instance, why can the model perform full retrieval but perform worse in partial retrieval? Why does the performance improve when CoT is used? Although it may be challenging to fully explain these observations, the paper would be more insightful if it included some hypotheses and experiments to verify these hypotheses.
>
> **Answer:** We probably didn’t explain this clear enough. For instance, the performance improves when CoT is used, because when using CoT, now (1) “A is B’s roommate” and (2) “B is hispanic” so (3) “A’s roommate is hispanic”. In this derivation, (1) and (2) are simply knowledge extraction (doable due to prior work 2309.14316), and (3) is simply in-context sentence copying. This makes the whole thing easy.
>
> In contrast, without CoT, the model has to directly say “A’s roommate is hispanic”, then it needs to use its internal states to derive who A’s roommate is before it can generate “hispanic”, and we showed that it cannot achieve so with good accuracy in this paper.
>
> We don't think **there is need for any “additional experiment” to verify this**, as our experiments are very complete: doing so without CoT is bad and doing so with CoT (equivalent to knowledge extraction) is good. In fact, this is exactly why we designed the manipulation tasks to be the simplest possible (like atomic), so that there’s no need to further divide the task into A and B to verify if A happened or B happened.
>
> The argument is similar in “partial retrieval”, when first stating the company city without the company name, this is a one-step knowledge classification without CoT and that’s why it fails.

---

> ### Author Response · Authors · 2024-11-28
> **Response to Reviewer afmD [2/2]**
>
> **Question:**
> > The paper explores limited variations in querying knowledge for LLMs, relying solely on the question forms + fine-tuning. But, other query methods, such as cloze-style prompting (with next-word prediction, which is closer to the pretraining objective), or zero-shot and few-shot prompting, could also be explored.
>
> **Answer:**  We do not think they need to be explored (except one, but we can tell you the result).
> * Zero/few-shot doesn’t directly make sense in our setting because our data is controlled and the queries (such as even a question-mark token) were never present in the pretrain data.
> * A more interesting question is what if we add also the queries to the pretrain data. This is called “mixed training” in this paper and we have done that in quite a few settings.
> * After mixed-training or finetuning, one can perhaps hope to use **few-shot** to improve accuracy. We have tried this and it doesn’t work — we can add this to the paper as a footnote. This should not be surprising, because 2-shot like “Joe Biden was born in an even month; Nancy Pelosi was born in an odd month; Donald Trump was born in an <bla>” can **only help the model to learn the format, but not the knowledge**. But our format is already sufficiently simple.
> * Cloze-style prompting seems irrelevant here because some of our tasks have single-token answers. For instance, asking the parity of birth month (even or odd), or the first name of a person with a given biography (usually a single token). In such cases, our finetune accuracy is the same as Cloze-style prompting.
>
> **Question:**
> > Regarding Weakness 1, what are the overall implications of the results? Could you clarify the broader conclusions/impacts that can be drawn from the current results?
>
> **Answer:** Implication? We have discovered some intrinsic issues with transformer-based language models. This already implies many data-related augmentations are necessary, and one has to do that before pre-training a language model (doing that at finetune stage is too late). This also provides some scientific testbed for future model/training method designs. Some follow-ups (including our own working ones) have exactly made use of this testbed to evaluate if the newly designed method can overcome such bottlenecks.
>
> **Question:**
> > For Weakness 2, are there any possible experiments that could provide further insights, or is this beyond the scope of the current study?
> > For Weakness 3, would conducting additional experiments with varied settings yield valuable insights, or are the current experiments sufficient to reach definitive conclusions?
>
> **Answer:**  As we mentioned above, we believe the experiments are complete, in the sense that **if it is a harder task of failure** (such as why the model cannot answer “Is A’s mother’s son’s birth year a leap year”) **then it is worth further dividing the task into simple ones** to see where exactly the failure is. But **we have already used the simplest possible manipulation tasks**. Perhaps the only one we can add, as we mentioned above, is that adding few-shot prompts doesn’t help either. We also tried to add “dot dot dot”, see Footnote 7, line 319-322, and it doesn’t help.

---

> > ### Comment · Reviewer_afmD · 2024-12-03
> >
> > I appreciate the author's discussion on my comments. It addresses my concerns, convincing me that my concerns are not very critical. I will raise my score.

---

### Meta-Review · Area_Chair_3R5i · 2024-12-19

**Metareview:**

**Summary**

This is a paper that confirms that LLMs are basically memorizers. Indeed, this paper examines the knowledge manipulation capabilities of LLMs, such as knowledge retrieval, classification, comparison, as well as inverse search. The expected discovery is that these models have substational limitations in performing even simple manipuation tasks such as identifying a person’s birth year if the information is not present in the specific form in training data.


**Strengths**

**Weaknesses**

- Manipulation tasks ar not clearly motivated.
- Models are fine-tuned. It is not clear if that is the reason for overmemorization.

**Final remarks**

The paper addresses an important known limitation [1,2] but it does not offer solutions.

**References**

* [1] Inbal Magar, Roy Schwartz, Data Contamination: From Memorization to Exploitation, 2022
* [2] Leonardo Ranaldi, Elena Sofia Ruzzetti, and Fabio Massimo Zanzotto. 2023. PreCog: Exploring the Relation between Memorization and Performance in Pre-trained Language Models.

**Additional Comments On Reviewer Discussion:**

The reviewers kept their scores during the rebuttal period.

---

### Decision · Program_Chairs · 2025-01-22

Accept (Poster)